

# Functional analysis of lncRNAs based on competitive endogenous RNA in tongue squamous cell carcinoma

Yidan Song[*], Yihua Pan[*] and Jun Liu

State Key Laboratory of Oral Diseases, National Clinical Research Center for Oral Diseases, Department of Orthodontics, West China Hospital of Stomatology, Sichuan University, Chengdu, China
[*] These authors contributed equally to this work.

Corresponding author
Jun Liu, junliu@scu.edu.cn

## ABSTRACT

**Backround**. Tongue squamous cell carcinoma (TSCC) is the most common malignant tumor in the oral cavity. An increasing number of studies have suggested that long noncoding RNA (lncRNA) plays an important role in the biological process of disease and is closely related to the occurrence and development of disease, including TSCC. Although many lncRNAs have been discovered, there remains a lack of research on the function and mechanism of lncRNAs. To better understand the clinical role and biological function of lncRNAs in TSCC, we conducted this study.

**Methods**. In this study, 162 tongue samples, including 147 TSCC samples and 15 normal control samples, were investigated and downloaded from The Cancer Genome Atlas (TCGA). We constructed a competitive endogenous RNA (ceRNA) regulatory network. Then, we investigated two lncRNAs as key lncRNAs using Kaplan–Meier curve analysis and constructed a key lncRNA-miRNA-mRNA subnetwork. Furthermore, gene set enrichment analysis (GSEA) was carried out on mRNAs in the subnetwork after multivariate survival analysis of the Cox proportional hazards regression model.

**Results**. The ceRNA regulatory network consists of six differentially expressed miRNAs (DEmiRNAs), 29 differentially expressed lncRNAs (DElncRNAs) and six differentially expressed mRNAs (DEmRNAs). Kaplan-Meier curve analysis of lncRNAs in the TSCC ceRNA regulatory network showed that only two lncRNAs, including LINC00261 and PART1, are correlated with the total survival time of TSCC patients. After we constructed the key lncRNA-miRNA -RNA sub network, the GSEA results showed that key lncRNA are mainly related to cytokines and the immune system. High expression levels of LINC00261 indicate a poor prognosis, while a high expression level of PART1 indicates a better prognosis.

## INTRODUCTION

Tongue squamous cell carcinoma (TSCC) is one of the most lethal malignancies of oral cancers, and it is the most common oral cancer (*Mannelli et al., 2018*; *Kamali et al., 2017*). TSCC is highly invasive and prone to recurrence and early metastasis, often endangering the lives of patients, and treatment is more difficult. Therefore, exploring the molecular mechanism of the occurrence and development of TSCC, studying the molecular markers

of TSCC and finding effective molecular therapeutic targets play an important role in improving the survival rate of patients with TSCC.

In recent years, a regulatory network composed of long noncoding RNAs (lncRNAs), microRNAs (miRNAs) and messenger RNAs (mRNAs) has gained interest in the study of molecular biological mechanisms involved in the process of tumor occurrence and progression. lncRNA is a noncoding RNA with a length greater than 200 nucleotides. Studies have shown that lncRNA plays an important role in many physiological activities, such as the dosage compensation effect, epigenetic regulation, cell cycle regulation and cell differentiation regulation (*Peng, Koirala & Mo, 2017*; *Koch, 2017*). In recent years, increasing evidence has shown that lncRNA has important potential application prospects in the treatment and diagnosis of malignant tumors and other human diseases (*Fan et al., 2018*; *Zhang et al., 2018c*).

miRNA is a kind of endogenous small RNA with a length of approximately 20–24 nucleotides that plays a variety of important regulatory roles in cells. miRNA is involved in the regulation of gene expression after transcription in cells. It can act on RNA and render it untranslatable, which leads to the silencing of corresponding genes (*Ye et al., 2008*). Researchers have shown that miRNA is directly related to many important physiological and biochemical processes and diseases, including tumor occurrence and metastasis (*Wang, Hu & Liu, 2015*; *Amirkhah et al., 2015*).

Some RNAs, pseudogenes, long-chain noncoding RNA (lncRNA) and cyclic RNA (circRNA) contain some binding sites for miRNAs, which can competently bind to the same miRNAs through miRNA response elements (MREs), thus eliminating or alleviating the inhibition of miRNAs on target genes and regulating the expression of target genes. This mechanism is referred to as the competitive endogenous RNA (ceRNA) hypothesis (*Qi et al., 2015*). An increasing number of studies have shown that in the process of tumorigenesis, lncRNA can be used as ceRNA, which can inhibit or promote tumorigenesis by competitive binding of the same tumorigenic or antitumorigenic miRNAs (*Karreth & Pandolfi, 2013*). For example, a study showed that lncRNA Gas5 acts as a ceRNA to regulate PTEN expression by sponging miR-222-3p in papillary thyroid carcinoma (*Zhang, Ye & Zhao, 2017*). In addition, it has been showed that lncRNA SNHG1 can antagonize the effect of miR-145a-5p on the downregulation of NUAK1 expression in nasopharyngeal carcinoma cells (*Lan & Liu, 2019*). In order to diagnose and treat oral tumors more accurately, more and more scholars are studying the mechanism of lncRNA in the development of oral tumors. One study explored the role of lncRNA in OSCC by conducting a series of analyses of data downloaded from GEO databases and conducting related experiments to verify the conclusions. Finally, the authors found that lncRNA H1, as a ceRNA, can promote the development of tumors by inhibiting the expression of miR-138 and EZH2, which is helpful for the treatment of OSCC (*Hong et al., 2018*). In addition, some scholars have found that lncRNA OIP5-AS1 is closely related to undifferentiated oral tumors, which can regulate downstream target genes. Moreover, lncRNA OIP5-AS1 can be transactivated by stemness-associated transcription factors in cancer, resulting in poor prognosis (*Arunkumar et al., 2018*).

There are an increasing number of studies on the construction of ceRNA networks for cancer research, such as gastric, lung, renal and kidney networks (*Xia et al., 2014*; *Wang et al., 2018*; *Qu et al., 2016*). However, until now, there has been nearly no research on lncRNAs and miRNAs related to TSCC based on high-throughput sequencing and a large-scale sample size. The purpose of this study was to analyze the expression of lncRNAs in TSCC and their correlation with clinical indicators based on the competitive endogenous RNA (ceRNA) theory and to explore the possible molecular mechanism of lncRNAs in TSCC by gene set enrichment analysis (GSEA). We aimed to provide new indicators for the early diagnosis and prognosis of TSCC and provide new targets for the treatment of TSCC.

## MATERIALS AND METHODS

### Selection of datasets and data collection

Gene expression data of TSCC were downloaded from The Cancer Genome Atlas (TCGA) (https://gdc-portal.nci.nih.gov/). TCGA is a project co-supervised by the National Cancer Institute and the National Human Genome Research Institute. It aims to apply high-throughput genome analysis technology to help people have a better understanding of cancer and improve the ability to prevent, diagnose and treat cancer. As the largest cancer gene information database at present, TCGA not only contains many cancer types, but also contains abundant multigroup data. We downloaded RNA-Seq V2 of tongue samples from TCGA, and this data is raw count, which comes from 162 tongue samples. Meanwhile, the downloaded miRNAs-seq came from 169 tongue samples.

### RNA sequence data processing

We downloaded all RNA sequences of 162 tongue samples from TCGA. In addition, we organized the downloaded data into gene expression matrix files and divided the 162 samples into 147 tumor samples and 15 normal control samples by the programming code written in Perl software. And we used the same method to convert the miRNA seq into miRNA matrix files and divided the 169 samples into 154 tumor samples and 15 normal control samples. And we also count the clinical data of the samples. The results are shown in the Table 1.

In this study, we mainly used programming code written in Perl software and R language to analyze the RNA data.

### Analysis of DEGs, DElncRNAs, and DEmiRNAs in TSCC

Ensembl (version 89; https://uswest.ensembl.org/info/website/archives/index.html) (*Aken et al., 2016*) is a software system capable of automatic annotation and maintenance of eukaryotic genomes. It is co-operated by the Wellcome Foundation of Sanger Institute and the European Institute of Bioinformatics, a division of the European Molecular Biology Laboratory. We used the Ensembl database to screen for RNA and lncRNA and exclude the RNA and lncRNA that do not exist in the database.

Before conducting differential expression analysis, we processed the data and deleted outliers samples. Using "WGCNA" package in R software (*R Core Team, 2018*), clustering

**Table 1  Clinical variables in TSCC samples.**

| Covariates | Type | Stat |
|---|---|---|
| fustat | dead | 58(39.5%) |
| fustat | alive | 89 (60.5%) |
| gender | male | 102(69.4%) |
| gender | female | 45(30.6%) |
| age | ≤65 | 69(46.9%) |
| age | >65 | 78(53.1%) |
| stage | I | 13(8.8 %) |
| stage | II | 23(15.6%) |
| stage | III | 32(21.8%) |
| stage | IV | 73(49.7%) |
| stage | NA | 6(4.1%) |

analysis was carried out on the expression data of mRNA, lncRNA and miRNA respectively, and outlier samples were deleted, the results are shown in supplemental files. Next, we obtained differentially expressed mRNAs (DEmRNAs), lncRNAs (DElncRNAs) and miRNAs (DEmiRNAs) using the "edgeR" package in R software. The $P$-value was obtained using the false discovery rate (FDR) to correct the statistical significance of multiple tests. We specified fold changes (log2 absolute) > 2 and an FDR adjusted to $P < 0.01$ as statistically significant.

## Construction of a ceRNA regulatory network

miRcode database is a software to predict microRNA targets. Currently, it covers a complete set of transcripts annotated by GENCODE, including 10,419 registered lncRNAs. When predicting the binding sites of miRNAs, we used the miRNA family defined by Target Scan v6. The miRNAs under the same family have the same binding site type. For the predicted region of the binding sites of miRNAs, the results are filtered and screened according to the gene type, the conservativeness of binding sites and the distribution of transcripts. First, the DElncRNAs and DEmiRNAs were matched in the miRcode database (*Jeggari, Marks & Larsson, 2012*). Unmatched lncRNAs and miRNAs were deleted. StarBase is used to annotate the suffix of miRNA. miRTarBase (*Chou et al., 2016*), miRDB (*Wong & Wang, 2015*) and TargetScan (*Fromm et al., 2015*) databases are all websites that can predict miRNA target genes. In order to make the results more accurate, we used the three databases to predict the target genes of DEmiRNAs. After that, the "Venn Diagram" package in R software was used to screen the target genes of DEmiRNAs by crossing the differentially expressed genes with the target genes, in order to prepare for the construction of ceRNA network. In order to measure the binding, a correlation analysis is necessary. We used the code in R software to detect the correlation between miRNAs and mRNAs, and between miRNAs and lncRNAs. $P < 0.05$ was considered statistically significant. The results of correlation analysis were provided in the supplemental files. Then, we obtained the ceRNA regulatory network of lncRNA-miRNA-mRNA and mapped it with Cytoscape software (http://www.cytoscape.org/).

## Survival analysis

We used the "survival" package in R software for survival analysis of all RNAs in the ceRNA network. The expression of lncRNAs, mRNAs and miRNAs correlated with overall survival was identified by the univariate Cox proportional hazards regression model. The expression of RNAs was sequenced from low to high. The first 50% of the samples were used as the low expression group, and the second 50% as the high expression group. Then, we used the log-rank test to compare the significant differences in the univariate analysis between subgroups for the overall survival rates. $P<0.05$ was considered statistically significant.

## Construction of the key lncRNA–miRNA–mRNA subnetwork

We used lncRNAs after survival analysis as key lncRNAs and extracted their linked mRNAs and miRNAs to reconstruct the subnetwork in Cytoscape software.

## Multivariate survival analysis of cox proportional risk regression model

The Cox proportional hazards regression model plays a key role in the clinical diagnosis of cancer and can be used to detect the risk of key genes. We first downloaded gene expression data from TCGA and then merged the data with Perl script to obtain relevant survival information. Then, mRNAs in the key lncRNA-miRNA-mRNA subnetwork were regarded as key genes. Next, we performed multivariate survival analysis of the key genes using the "survival" package in R software, and the risk level of key genes was obtained. We provided the multi-variate analysis in the Supplemental Files.

## Gene Set Enrichment Analysis (GSEA)

GSEA connects expression profile chip data with biological significance through statistical analysis, which enables researchers to interpret chip results more easily and reasonably. To further explore the role of the ceRNA regulatory network in the occurrence and development of TSCC, after multivariate analysis, the key genes were divided into two groups according to the risk level as follows: h represents the high gene expression group and l represents the low gene expression group. Then, the results were imported into the GSEA software. This analysis used two gene sets as follows: the KEGG gene set and the gene ontology gene set, including biological processes, cellular components and molecular functions. FDRs (false discovery rates) <25% for the gene sets indicated significantly enriched gene sets using the GSEA software. In the GSEA results, the gene set with FDR < 25% is the enrichment set by default. Because it is possible to generate meaningful hypotheses from these functional genes to facilitate further research. In most cases, the choice of FDR less than 25% is appropriate.

# RESULTS

## DERNAs between primary tumor and control samples

Before the differential expression analysis, the clustering analysis were performed to check the outlier samples. Finally, one cancer sample was deleted from the mRNA data, four control samples were deleted from the lncRNA data, and one control sample was deleted from the miRNA data, the results are shown in Supplemental Files. Based on
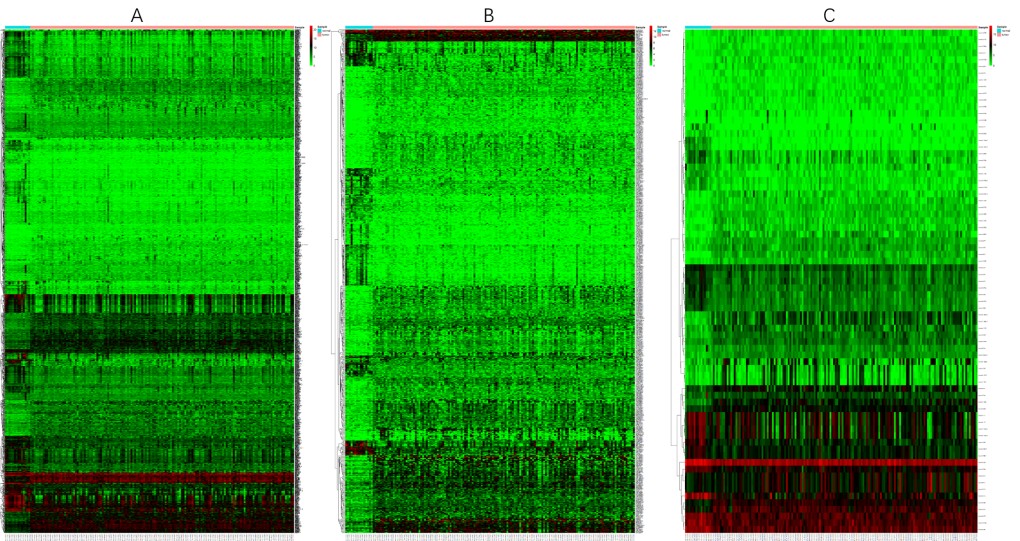

**Figure 1  Heat maps of DERNAs.** (A) Heat map of DEmRNAs. (B) Heat map of DElncRNAs. (C) Heat map of DEmiRNAs. The red and green colors indicate higher expression levels and lower expression levels, respectively. The rows represent the DERNAs and the columns represent the samples.

the data downloaded from TCGA, using a logFC >2 and an FDR <0.01 as the cut-off criterion, we screened 1,590 DEmRNAs (Fig. 1), including 608 upregulated mRNAs and 982 downregulated mRNAs. Similarly, 627 DElncRNAs (Fig. 1) and 75 DEmiRNAs (Fig. 1) were obtained, including 401 upregulated and 226 downregulated lncRNAs and 41 upregulated and 34 downregulated miRNAs. Tables 2–4 show the first 25 upregulated RNAs and 25 downregulated RNAs and their corresponding logFC values, *P*-values, and FDR values. The complete list of DERNAs is provided in supplemental files.

## Construction of the ceRNA regulatory network

To better understand the role of DElncRNAs in TSCC and further elaborate on the interaction between DElncRNAs and DEmiRNAs, a ceRNA regulatory network map related to lncRNA-miRNA-mRNA was constructed using 627 DElncRNAs and 75 DEmiRNAs obtained from the abovementioned studies. In the ceRNA regulatory network of TSCC, the interaction between 627 DEIncRNAs and 75 DERNAs was predicted based on human gene data in the downloaded microcode database using the Perl program. A total of 45 pairs of interacting lncRNAs and microRNAs were obtained. According to the microTarBase, TargetScan and microRNADB databases, 99 DEmiRNAs targeting mRNAs were searched. Then, eight differentially expressed and targeted mRNAs were screened out by comparing the targeted mRNAs with TSCC DEmRNAs from TCGA. Cytoscape was used to construct a ceRNA regulatory network including six DEmiRNAs, 29 DElncRNAs and six DEmRNAs in TSCC based on the abovementioned RNAs (Fig. 2).

Song et al. (2019), *PeerJ*, DOI 10.7717/peerj.6991

**Table 2  Top 25 differentially expressed mRNAs in TSCC samples.**

| Top 25 up-regulated mRNAs | | | | | | Top 25 down-regulated mRNAs | | | | | |
|---|---|---|---|---|---|---|---|---|---|---|---|
| mRNA | Description | logFC | logCPM | *P* Value | FDR | mRNA | Description | logFC | logCPM | *P* Value | FDR |
| HOXC8 | homeobox C8 | 5.283727 | 0.312936 | 4.53E–19 | 2.79E–17 | PRB3 | proline rich protein BstNI subfamily 3 | −9.70388 | 4.872374 | 7.81E–98 | 1.40E–93 |
| BMP1 | bone morphogenetic protein 1 | 2.198072 | 6.638635 | 3.16E—18 | 1.81E–16 | PRH1 | proline rich protein HaeIII subfamily 1 | −7.72276 | 2.560288 | 3.04E–94 | 2.72E–90 |
| MFAP2 | microfibril associated protein 2 | 3.188861 | 6.042071 | 5.08E–18 | 2.81E–16 | LCN1 | lipocalin 1 | −11.3997 | 8.089483 | 3.38E–79 | 2.02E–75 |
| PLAU | plasminogen activator, urokinase | 3.005871 | 8.32712 | 8.47E–18 | 4.60E–16 | PRR4 | proline rich 4 | −8.31566 | 5.865882 | 2.35E–76 | 1.05E–72 |
| C1QTNF6 | C1q and TNF related 6 | 3.264884 | 5.673275 | 4.53E–17 | 2.27E–15 | OMG | oligodendrocyte myelin glycoprotein | −4.8649 | −0.11353 | 1.75E–73 | 6.28E–70 |
| CD276 | CD276 molecule | 2.09334 | 7.165463 | 7.14E–17 | 3.48E–15 | RSPH4A | radial spoke head component 4A | −5.17019 | 0.179478 | 1.19E–65 | 3.57E–62 |
| SERPINH1 | serpin family H member 1 | 2.511446 | 8.51167 | 7.17E–17 | 3.48E–15 | KRT85 | keratin 85 | −7.19377 | 0.580109 | 2.80E–62 | 7.16E–59 |
| COL13A1 | serpin family H member 1 | 2.529188 | 2.474665 | 1.74E–16 | 8.05E–15 | RSPH1 | radial spoke head component 1 | −5.36305 | 0.850217 | 1.37E–61 | 3.06E–58 |
| HOXC6 | homeobox C6 | 4.461024 | 1.226607 | 2.93E–16 | 1.34E–14 | VWA3B | von Willebrand factor A domain containing 3B | −4.96649 | 0.321375 | 1.12E–59 | 2.24E–56 |
| TGFBI | transforming growth factor beta induced | 3.851308 | 10.07835 | 1.11E–15 | 4.78E–14 | ZG16B | zymogen granule protein 16B | −8.05015 | 7.996396 | 1.76E–57 | 3.16E–54 |
| HOXC4 | homeobox C4 | 3.777635 | 1.189556 | 1.40E–15 | 6.00E–14 | C6orf58 | chromosome 6 open reading frame 58 | −8.42169 | 4.31457 | 1.14E–56 | 1.86E–53 |
| HSD17B6 | hydroxysteroid 17-beta dehydrogenase 6 | 2.405452 | 1.277905 | 5.23E–15 | 2.07E–13 | ZMYND10 | zinc finger MYND-type containing 10 | −5.1136 | 1.208837 | 6.75E–54 | 1.01E–50 |
| LAMC2 | laminin subunit gamma 2 | 3.836815 | 10.61456 | 5.85E–15 | 2.29E–13 | DNAH9 | dynein axonemal heavy chain 9 | −4.73983 | 1.343683 | 2.57E–53 | 3.54E–50 |
| GRIN2D | glutamate ionotropic receptor NMDA type subunit 2D | 4.331032 | 3.546628 | 9.64E–15 | 3.69E–13 | C9orf24 | chromosome 9 open reading frame 24 | −4.84518 | −0.09058 | 4.11E–53 | 5.26E–50 |
| PDPN | podoplanin | 2.695333 | 7.36453 | 2.52E–14 | 9.09E–13 | CFAP43 | cilia and flagella associated protein 43 | −4.37867 | 0.383593 | 4.26E–52 | 5.09E–49 |
| IL11 | interleukin 11 | 3.881345 | 3.0041 | 3.16E–14 | 1.12E–12 | ECT2L | epithelial cell transforming 2 like | −4.08998 | −1.00566 | 2.97E–51 | 3.33E–48 |
| CA9 | carbonic anhydrase 9 | 5.962956 | 5.020682 | 3.36E–14 | 1.18E–12 | PIFO | primary cilia formation | −4.33282 | 0.834553 | 9.28E–51 | 9.51E–48 |
| P3H1 | prolyl 3-hydroxylase 1 | 2.182197 | 5.66245 | 4.70E–14 | 1.63E–12 | FAM166B | family with sequence similarity 166 member B | −3.57818 | −0.50684 | 9.55E–51 | 9.51E–48 |
| LBX2 | ladybird homeobox 2 | 2.707215 | −0.43299 | 1.27E–13 | 4.15E–12 | LRRC46 | leucine rich repeat containing 46 | −4.07671 | 0.796168 | 7.59E–50 | 7.16E–47 |
| ZNF114 | zinc finger protein 114 | 3.990454 | 2.237109 | 1.33E–13 | 4.32E–12 | TEKT1 | tektin 1 | −5.7258 | 0.094967 | 4.01E–49 | 3.59E–46 |
| SNX10 | sorting nexin 10 | 2.139152 | 4.011967 | 1.34E–13 | 4.32E–12 | WDR38 | WD repeat domain 38 | −5.66533 | −0.23915 | 1.25E–48 | 1.07E–45 |
| DNMT3B | DNA methyltransferase 3 beta | 2.4099 | 3.445032 | 1.42E–13 | 4.57E–12 | PRB4 | proline rich protein BstNI subfamily 4 | −9.41497 | −0.08209 | 3.81E–47 | 3.10E–44 |
| TMEM132A | transmembrane protein 132A | 2.071182 | 7.053422 | 1.75E–13 | 5.51E–12 | EFHB | EF-hand domain family member B | −4.48575 | −0.84504 | 5.50E–47 | 4.28E–44 |
| HOXC9 | homeobox C9 | 3.908623 | 0.666057 | 2.46E–13 | 7.54E–12 | ERICH3 | glutamate rich 3 | −6.91452 | 0.213528 | 1.66E–46 | 1.24E–43 |
| ARTN | artemin | 3.278099 | 3.346497 | 2.76E–13 | 8.34E–12 | LPO | lactoperoxidase | −6.76434 | 1.46561 | 3.06E–46 | 2.20E–43 |

**Table 3 Top 25 differentially expressed lncRNAs in TSCC samples.**

| Top 25 up-regulated lncRNAs | | | | | | Top 25 down-regulated lncRNAs | | | | |
| lncRNA | Description | logFC | logCPM | PValue | FDR | lncRNA | Description | logFC | logCPM | PValue | FDR |
| --- | --- | --- | --- | --- | --- | --- | --- | --- | --- | --- | --- |
| LINC02081 | S-phase cancer associated transcript 1 | 6.55907 | 8.288038 | 1.91E−17 | 6.25E−15 | LINC01482 | Long intergenic non-protein coding RNA 1482 | −4.74187 | 5.152461 | 1.46E−46 | 1.05E−42 |
| HOXC-AS2 | HOXC cluster antisense RNA 2 | 4.769257 | 6.822429 | 1.98E−16 | 6.23E−14 | LINC01829 | long intergenic non-protein coding RNA 1829 | −6.36522 | 6.723502 | 3.81E−39 | 1.38E−35 |
| ZFPM2-AS1 | ZFPM2 antisense RNA 1 | 6.665231 | 8.995423 | 6.58E−16 | 1.83E−13 | AC027130.1 | Antisense | −5.97661 | 4.460428 | 1.08E−36 | 2.61E−33 |
| GSEC | G-quadruplex forming sequence containing lncRNA | 3.169551 | 8.755699 | 2.53E−15 | 5.70E−13 | LINC02160 | long intergenic non-protein coding RNA 2160 | −6.54927 | 4.019298 | 2.85E−35 | 5.15E−32 |
| AL358334.2 | novel transcript | 5.390387 | 8.490721 | 3.08E−15 | 6.73E−13 | RMST | rhabdomyosarcoma 2 associated transcript | −5.26688 | 3.845394 | 2.79E−32 | 4.03E−29 |
| AC114956.2 | novel transcript | 4.310233 | 8.004771 | 2.70E−14 | 5.13E−12 | AC010425.1 | novel transcript | −5.48585 | 3.476603 | 1.74E−26 | 2.10E−23 |
| FOXD2-AS1 | adjacent opposite strand RNA 1 | 2.583099 | 8.660159 | 1.58E−13 | 2.60E−11 | LINC01798 | long intergenic non-protein coding RNA 1798 | −3.58562 | 4.324626 | 8.21E−26 | 8.46E−23 |
| AC114956.1 | novel transcript | 4.215066 | 6.515797 | 5.15E−13 | 7.91E−11 | LINC01797 | long intergenic non-protein coding RNA 1797 | −6.20978 | 3.390637 | 1.23E−24 | 1.11E−21 |
| HOXA10-AS | HOXA10 antisense RNA | 5.742475 | 5.845017 | 1.66E−12 | 2.36E−10 | FOXCUT | FOXC1 upstream transcript | −4.07718 | 5.643834 | 6.56E−24 | 5.26E−21 |
| LINC02577 | long intergenic non-protein coding RNA 2577 | 6.002243 | 9.022875 | 1.04E−11 | 1.20E−09 | SALRNA1 | senescence associated long non-coding RNA 1 | −3.08355 | 5.220263 | 1.34E−20 | 9.64E−18 |
| LINC00460 | long intergenic non-protein coding RNA 460 | 6.495702 | 7.974214 | 2.49E−11 | 2.68E−09 | AC005165.1 | uncharacterized LOC646588 | −4.48757 | 6.872983 | 4.67E−20 | 3.06E−17 |
| AC134312.5 | novel transcript | 4.486535 | 7.851042 | 4.56E−11 | 4.28E−09 | AL451062.1 | Antisense | −4.06772 | 5.687606 | 1.08E−19 | 6.49E−17 |
| LBX2-AS1 | LBX2 antisense RNA 1 | 2.21821 | 9.536711 | 6.89E−11 | 6.14E−09 | LINC00844 | long intergenic non-protein coding RNA 844 | −4.33944 | 3.690738 | 2.61E−19 | 1.45E−16 |
| AC005041.3 | novel transcript, antisense to LBX2 | 2.306235 | 5.550463 | 8.96E−11 | 7.79E−09 | LINC01405 | long intergenic non-protein coding RNA 1405 | −4.17211 | 8.680984 | 2.83E−19 | 1.45E−16 |
| LINC01633 | long intergenic non-protein coding RNA 1633 | 5.183034 | 4.912757 | 1.17E−10 | 9.92E−09 | AC116345.1 | novel transcript | −3.55372 | 5.032557 | 3.01E−19 | 1.45E−16 |
| AL513123.1 | novel transcript | 4.39058 | 5.896268 | 1.55E−10 | 1.26E−08 | AL359715.1 | novel transcript | −2.78923 | 5.706472 | 4.44E−19 | 2.00E−16 |
| AL139147.1 | novel transcript, antisense to SGIP1 | 7.261896 | 5.873938 | 1.82E−10 | 1.44E−08 | AC009041.2 | novel transcript | −2.81453 | 7.784602 | 2.09E−18 | 8.87E−16 |
| U62317.1 | uncharacterized LOC105373098 | 3.307612 | 10.06709 | 2.88E−10 | 2.19E−08 | NKAIN3-IT1 | NKAIN3 intronic transcript | −4.26922 | 5.38921 | 3.62E−18 | 1.40E−15 |
| HOTAIR | HOX transcript antisense RNA | 4.576517 | 6.201958 | 3.63E−10 | 2.73E−08 | HSD52 | uncharacterized LOC729467 | −3.84183 | 3.899436 | 3.69E−18 | 1.40E−15 |
| LINC01615 | long intergenic non-protein coding RNA 1615 | 4.411259 | 9.631277 | 4.36E−10 | 3.21E−08 | LINC02343 | long intergenic non-protein coding RNA 2343 | −5.54729 | 3.466708 | 6.86E−18 | 2.48E−15 |
| SLC12A5-AS1 | SLC12A5 and MMP9 antisense RNA 1 | 3.954435 | 6.183039 | 4.91E−10 | 3.54E−08 | AL161668.4 | novel transcript | −3.7574 | 4.738385 | 7.66E−18 | 2.63E−15 |
| AC113346.1 | novel transcript | 5.684313 | 5.792491 | 5.03E−10 | 3.60E−08 | SOX9-AS1 | SOX9 antisense RNA 1 | −2.54064 | 7.026463 | 2.56E−16 | 7.70E−14 |
| LINC00941 | long intergenic non-protein coding RNA 941 | 4.33015 | 8.913483 | 5.20E−10 | 3.68E−08 | ZNF710-AS1 | ZNF710 antisense RNA 1 | −2.32401 | 9.970743 | 6.09E−16 | 1.76E−13 |
| RNF144A-AS1 | RNF144A antisense RNA 1 | 3.391581 | 7.876758 | 7.03E−10 | 4.83E−08 | DRAIC | downregulated RNA in cancer, inhibitor of cell invasion and migration | −2.84397 | 5.017242 | 7.74E−16 | 2.07E−13 |
| LINC01614 | long intergenic non-protein coding RNA 1614 | 7.771486 | 8.166619 | 1.24E−09 | 8.26E−08 | AC002401.3 | novel transcript, antisense to PDK2 | −3.1933 | 3.763892 | 1.10E−15 | 2.84E−13 |

**Table 4  Top 25 differentially expressed miRNAs in TSCC samples.**

| miRNA | Top 25 up-regulated miRNAs logFC | logCPM | PValue | FDR | miRNA | Top 25 down-regulated miRNAs logFC | logCPM | PValue | FDR |
|---|---|---|---|---|---|---|---|---|---|
| hsa-mir-615 | 5.730799 | 1.478439 | 6.70E−17 | 2.22E−15 | hsa-mir-299 | −2.99771 | 3.863924 | 1.11E−32 | 6.98E−30 |
| hsa-mir-455 | 2.538941 | 8.844984 | 5.06E−15 | 1.38E−13 | hsa-mir-135a-2 | −5.11378 | 0.187823 | 3.80E−32 | 1.19E−29 |
| hsa-mir-196b | 4.394713 | 7.097162 | 1.72E−13 | 3.72E−12 | hsa-mir-381 | −3.59226 | 7.848088 | 4.55E−31 | 9.53E−29 |
| hsa-mir-196a-2 | 4.970325 | 5.132245 | 2.79E−12 | 5.17E−11 | hsa-mir-135a-1 | −4.56572 | −0.16831 | 1.59E−28 | 2.51E−26 |
| hsa-mir-503 | 3.184672 | 3.776945 | 1.55E−11 | 2.57E−10 | hsa-mir-30a | −2.10026 | 13.63923 | 3.18E−23 | 4.01E−21 |
| hsa-mir-196a-1 | 4.906809 | 4.980926 | 1.94E−11 | 2.97E−10 | hsa-mir-885 | −4.10373 | 1.650439 | 4.81E−22 | 5.04E−20 |
| hsa-mir-224 | 2.141737 | 7.121961 | 9.49E−10 | 1.15E−08 | hsa-mir-378c | −2.49589 | 4.056665 | 9.94E−22 | 8.93E−20 |
| hsa-mir-450a-1 | 2.045314 | 2.394319 | 1.39E−09 | 1.62E−08 | hsa-mir-411 | −2.72564 | 4.401595 | 1.26E−21 | 9.90E−20 |
| hsa-mir-877 | 2.435612 | 1.584991 | 2.04E−09 | 2.33E−08 | hsa-mir-34c | −2.75873 | 5.722292 | 3.57E−21 | 2.50E−19 |
| hsa-mir-1910 | 3.99605 | 0.658871 | 2.55E−09 | 2.82E−08 | hsa-mir-375 | −3.96099 | 11.45332 | 2.19E−19 | 1.15E−17 |
| hsa-mir-301a | 2.09408 | 3.746788 | 3.37E−08 | 3.17E−07 | hsa-mir-378a | −2.24997 | 10.81757 | 6.76E−19 | 3.04E−17 |
| hsa-mir-4652 | 4.667714 | 1.65468 | 6.24E−08 | 5.45E−07 | hsa-mir-376c | −2.27353 | 3.501245 | 1.03E−18 | 4.31E−17 |
| hsa-mir-1305 | 3.364084 | 0.561589 | 1.34E−07 | 1.11E−06 | hsa-mir-495 | −2.18082 | 3.324704 | 3.04E−18 | 1.19E−16 |
| hsa-mir-210 | 2.331449 | 10.02408 | 3.90E−07 | 2.89E−06 | hsa-mir-486-2 | −2.33925 | 7.190098 | 9.14E−18 | 3.35E−16 |
| hsa-mir-7705 | 2.102799 | 0.65585 | 6.32E−07 | 4.51E−06 | hsa-mir-486-1 | −2.35431 | 7.205133 | 9.59E−18 | 3.35E−16 |
| hsa-mir-937 | 2.549417 | 2.138107 | 8.21E−07 | 5.80E−06 | hsa-mir-29c | −2.08372 | 11.16676 | 1.91E−16 | 6.01E−15 |
| hsa-mir-7854 | 2.596544 | −0.02223 | 1.05E−06 | 7.28E−06 | hsa-mir-378d-2 | −2.58036 | 0.400949 | 2.85E−16 | 8.52E−15 |
| hsa-mir-3940 | 2.188397 | 0.789722 | 1.10E−06 | 7.53E−06 | hsa-mir-34b | −2.52533 | 3.09759 | 3.37E−15 | 9.63E−14 |
| hsa-mir-301b | 2.409846 | 1.003252 | 1.36E−06 | 9.09E−06 | hsa-mir-337 | −2.10921 | 5.058 | 9.56E−15 | 2.51E−13 |
| hsa-mir-545 | 2.274835 | −0.21026 | 1.89E−06 | 1.22E−05 | hsa-mir-379 | −2.01836 | 10.57229 | 1.61E−14 | 4.06E−13 |
| hsa-mir-1293 | 3.020534 | 4.296348 | 2.21E−06 | 1.39E−05 | hsa-mir-1258 | −2.5535 | 0.159542 | 2.30E−14 | 5.57E−13 |
| hsa-mir-31 | 2.620164 | 7.436808 | 3.09E−06 | 1.85E−05 | hsa-mir-410 | −2.26437 | 4.466849 | 3.02E−14 | 7.04E−13 |
| hsa-mir-4713 | 2.832803 | −0.29741 | 9.36E−06 | 5.26E−05 | hsa-mir-378d-1 | −2.55416 | 0.146215 | 9.49E−12 | 1.66E−10 |
| hsa-mir-4658 | 3.809692 | −0.47128 | 1.54E−05 | 8.33E−05 | hsa-mir-499a | −2.97542 | 2.103145 | 2.54E−11 | 3.80E−10 |
| hsa-mir-548f-1 | 4.284969 | 0.072804 | 1.87E−05 | 9.87E−05 | hsa-mir-99a | −2.01938 | 9.003363 | 5.18E−11 | 7.24E−10 |

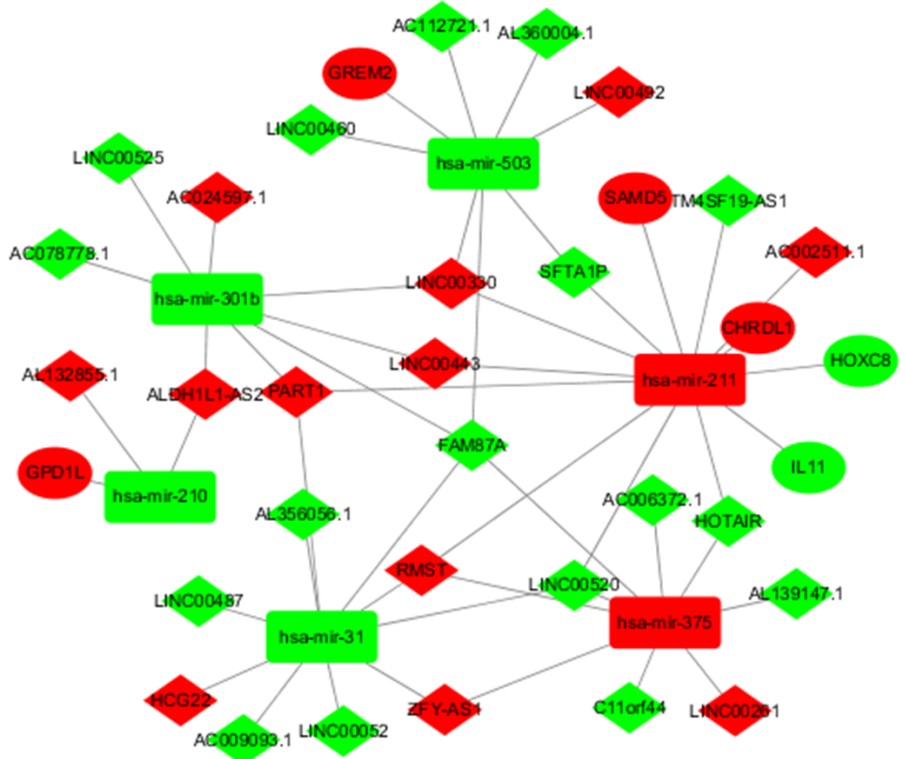

**Figure 2  DElncRNA—mediated ceRNA regulatory network in tongue squamous cellcarcinoma.** The nodes highlighted in red indicate downregulated expression, and the nodes highlighted in green indicate upregulated expression. lncRNAs, miRNAs and mRNAs are represented by diamonds, rounded rectangles, and ellipses, respectively.

## Correlation analysis of survival in the ceRNA network

To further determine whether the expression of genes in the regulatory network is related to the prognosis of TSCC patients, Kaplan–Meier curve analysis of lncRNAs in the TSCC ceRNA regulatory network showed that only two lncRNAs, including LINC00261 and PART1, were correlated with the overall survival time of TSCC patients ($P < 0.05$), as shown in Fig. 3. Moreover, the expression levels of LINC00261 were negatively correlated with the overall survival rate, while the expression levels of PART1 were positively correlated with the overall survival rate.

## The key lncRNA-miRNA-mRNA sub-network

LncRNAs and mRNAs appear in the coexpression patterns of ceRNA. Thus, we used two lncRNAs, including LINC00261 and PART1, as the key lncRNAs. We extracted their linked miRNAs and mRNAs and reconstructed a new subnetwork using Cytoscape software. As shown in Fig. 4, The lncRNA PART1-miRNA-mRNA subnetwork was composed of one lncRNA node, three miRNA nodes, and four mRNA nodes.

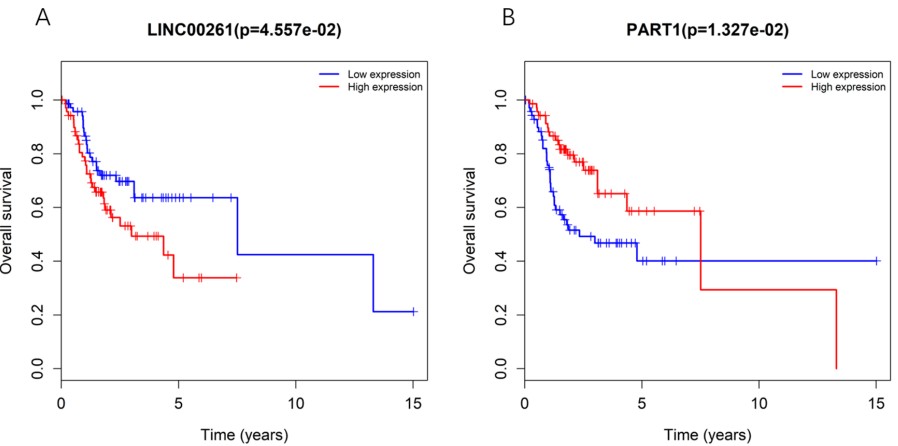

**Figure 3** **Kaplan-Meiercurve analysis of DElncRNA (A: *LINC00261* and B: *PART1*) inTSCC patients.**
Blue represents low expression and red represents high expression.

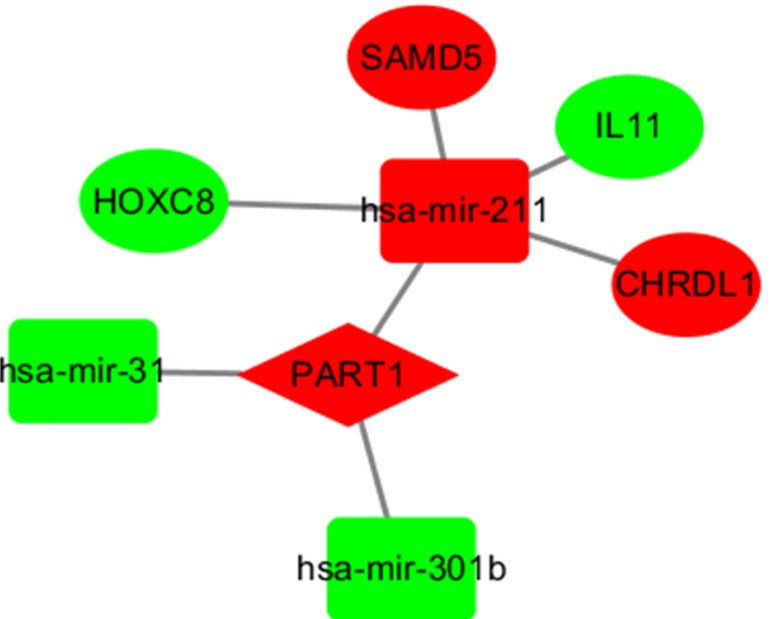

**Figure 4** **The subnetwork of PART1.** The nodes highlighted in red indicate downregulated expression, and the nodes highlighted in green indicate upregulated expression. lncRNAs, miRNAs and mRNAs are represented by diamonds, rounded rectangles, and ellipses, respectively.

## Functional prediction of lncRNAs based on the key lncRNA-miRNA-mRNA subnetwork

In the network, one or more mRNAs are centered on a lncRNA and connected with it. We can better speculate the function of lncRNA according to the function of the mRNA linked to the lncRNA. Therefore, after multifactor analysis of the mRNAs in the constructed subnetwork, the subnetwork was divided into a high-expression group and
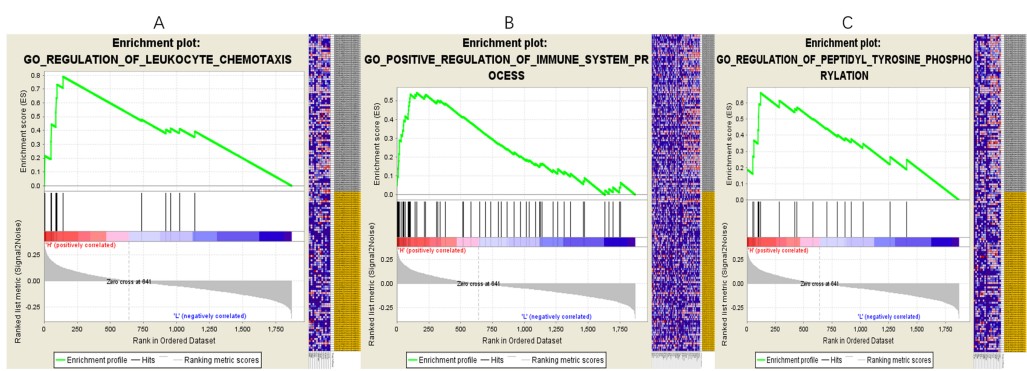

**Figure 5** **Gene setenrichment analysis of the subnetwork.** Gene set enrichment analysis (GSEA) of mRNAs in the subnetwork demonstrated that PART1 is enriched in genes that participate in leukocyte chemotaxis (A), immune system process (B) and peptidyl tyrosine phosphorylation (C).

a low-expression group according to the risk value, and GSEA analysis was then carried out to explore the biological pathways involved in lncRNA. The GSEA results showed that in the high-expression group of PART1(Fig. 5), six significantly enriched gene sets were screened, including one leukocyte migration gene set, two leukocyte chemotaxis gene sets, one immune system process gene set, one innate immune response gene set and one tyrosine peptide phosphorylation gene set.

## DISCUSSION

In recent decades, TSCC is the most common oral squamous cell carcinoma, and its treatment is difficult. Currently, basic research and clinical treatment of tongue cancer are the focuses of head and neck cancer surgery. The overall clinical treatment of tongue cancer has improved in recent decades. The 5-year survival rate is 50–60% (*Sano & Myers, 2007*; *Torre et al., 2015*), but it has been stagnant. Neoadjuvant therapies, such as biotherapy and targeted therapy, have been gradually applied in clinical practice. It has been shown that ceRNAs alter gene expression through a mechanism mediated by microRNAs, thereby affecting cell function, which may lead to cancer (*Qi et al., 2015*). However, only a few studies have been performed on ceRNA in TSCC.

More than 10,000 kinds of lncRNAs have been shown to have potential ceRNA characteristics (*Tay, Rinn & Pandolfi, 2014*). In recent years, an increasing number of studies have confirmed that lncRNAs, as a competitive platform for miRNAs and mRNAs, is involved in the regulation of the cell cycle and cell death in various malignant tumors, such as breast, gastric, liver, lung and esophageal cancers (*Luan et al., 2017*; *Zhang et al., 2018a*; *Wang et al., 2017a*; *Jiao et al., 2016*). LncRNA effects the invasion and metastasis of tumors, thus playing an important role in the occurrence and development of tumors. In the study of TSCC, many lncRNAs have been shown to be associated with the occurrence, development and prognosis of TSCC. In a study published in 2018, a high level of lncRNA KCNQ1OT1 expression was shown to be closely correlated with poor prognosis in TSCC. Furthermore, KCNQ1OT1 was shown to promote TSCC proliferation (*Zhang et al., 2018b*).

Another study showed that lncRNA MALAT1 expression is upregulated in TSCC tissues by investigating the expression and function of MALAT1 in TSCC tissues and cells, and MALAT1 was shown to be related to cervical lymph node metastasis. Finally, MALAT1 was shown to induce cell migration, invasion and EMT and inhibit apoptosis by modulating the Wnt/ β-catenin signaling pathway (*Liang et al., 2017*). These results may provide new insights for the treatment of TSCC.

Therefore, to study the functional roles of lncRNAs as ceRNA in the occurrence and development of TSCC and the regulatory mechanism of lncRNAs, we used data downloaded from TCGA to create a lncRNA-miRNA-mRNA network and carried out a series of analyses, which is very relevant for the treatment of TSCC. In our study, two lncRNAs, including LINC00261 and PART1, were shown to be closely related to overall survival. Patients with high expression levels of LINC00261 had a poor prognosis, while PART1 had the opposite effect. The results of other studies are different from this study. A study (*Yu et al., 2017*) showed that patients with low expression levels of LINC00261 had a higher recurrence rate than those with high expression levels of LINC00261 in gastric cancer, suggesting that patients with low expression levels of LINC00261 have a poor prognosis. We hypothesize that the results are different because of the small number of samples in the present study.

Currently, there are few studies on the two kinds of key lncRNAs in TSCC, but many scholars have indicated that these key lncRNAs have an impact on the occurrence and development of other diseases. LINC00261 plays an important role in gastric, lung and colon cancers (*Liu, Xiao & Xu, 2017*; *Wang et al., 2017c*). Many studies have proven that LINC00261 can inhibit cell proliferation and invasion, thus promoting the development of cancer (*Sha et al., 2017*; *Wang et al., 2017b*). There are fewer studies on PART1 than LNC00261, and only a few studies have investigated its role in prostate cancer and esophageal squamous cell carcinoma (*Sun et al., 2018*; *Kang et al., 2018*). A 2018 study showed that lncRNA PART1 modulates the toll-like receptor pathway to influence cell proliferation and apoptosis in prostate cancer cells. PART1 can be used as a novel biomarker and target in the treatment of prostate cancer (*Sun et al., 2018*).

The results of enrichment analysis of the lncRNA-miRNA-mRNA subnetwork showed that PART1 is mainly related to cytokines and the immune system and could regulate the immune response and migration of cytokines, thus affecting the occurrence and development of tumors. However, little research has been performed on the effects of these two lncRNAs on immunity and cytokines.

## CONCLUSIONS

Based on the ceRNA theory, we constructed a ceRNA regulatory network, which for the first time enabled an overall perspective and analysis of the lncRNA-associated ceRNA-mediated genes in the development of TSCC at a system-wide level. Our research showed that lncRNAs affect the development of TSCC. Furthermore, the constructed key lncRNA-miRNA-mRNA subnetwork also showed that PART1 can provide new indicators for the early diagnosis and prognosis of TSCC and provide new targets for its treatment. However, the molecular mechanism of the two lncRNAs in TSCC remains unclear. Although the

GSEA results provided some research directions, further experiments are needed to verify it.

## ACKNOWLEDGEMENTS

We would like to express our sincere thanks to all those who have lent us hands in the course of our writing this paper. In particular, we would like to thank Dr. Jun Liu for his many suggestions and efforts to improve the research deficiencies. With his help, we accomplished it more smoothly.

### Funding

This work was supported by the National Natural Science Foundation of China (No.81870743 and No.81470722) and the Creative Spark Fund of Sichuan University (No.2018SCUH0007). The funders had no role in study design, data collection and analysis, decision to publish, or preparation of the manuscript.

### Grant Disclosures

The following grant information was disclosed by the authors:
National Natural Science Foundation of China: 81870743, 81470722.
Creative Spark Fund of Sichuan University: 2018SCUH0007.

### Competing Interests

The authors declare there are no competing interests.

### Author Contributions

- Yidan Song conceived and designed the experiments, performed the experiments, analyzed the data, contributed reagents/materials/analysis tools, prepared figures and/or tables, authored or reviewed drafts of the paper, approved the final draft.
- Yihua Pan and Jun Liu conceived and designed the experiments, performed the experiments, analyzed the data, contributed reagents/materials/analysis tools, prepared figures and/or tables, authored or reviewed drafts of the paper.

### Data Availability

Gene expression data of tongue squamous cell carcinoma (TSCC) were downloaded from The Cancer Genome Atlas (TCGA) (https://gdc-portal.nci.nih.gov/). In this study, 162 tongue samples, including 147 TSCC samples and 15 normal control samples, were investigated.

Raw data is available in the Supplemental Files.

### Supplemental Information

Supplemental information for this article can be found online at http://dx.doi.org/10.7717/peerj.6991#supplemental-information.

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
