# Peer review of "Functional analysis of lncRNAs based on competitive endogenous RNA in tongue squamous cell carcinoma"

_PeerJ, doi:10.7717/peerj.6991_

## Round 0.1 · original submission · Major Revisions

Technical details require better explanation, additional validations of the methods are required. Several sections of the manuscript need to be streamlined and re-written for clarity. The reader will benefit from description of the analyzed dataset.

Reviewer 1 ·

Basic reporting

Some sections of the manuscript, especially the methods section lacks clarity. Figure 1, 2, and 3 can be combined into a single figure with multiple panels. The volcano plots can be moved to supplementary figures. Figure 4 is redundant and can be moved into supplementary. Also, Figure 7, 8 and 9 can be combined into a single figure with multiple panels.

Experimental design

The study aims to find out the roles of lncRNAs as part of putative competing endogenous RNA network in tongue squamous cell carcinoma. While the overall design of the study is well defined and the results support the conclusions, I would like to point out some aspects where the manuscript can be more clear.
1. In the Methods section, proper description of the TCGA dataset should be given. Instead of "RNA sequences", the authors should mention whether they used RNA-Seq v2 and/or miRNA-seq dataset from TCGA, and in which format the RNA-seq data was downloaded; was it raw count or RSEM or FPKM? Also, the authors mentioned "Before conducting differential expression analysis, we processed the data and deleted the RNA and miRNA data that were not expressed"; was there a cut-off for expression or number of samples (e.g. remove genes with < 1 FPKM in 90% of samples)?
2. They mentioned about multi-variate analysis with mRNAs in the ceRNA network. But no figures or tables are included that shows the result of the multi-variate analysis. How was the risk scores and what was the significance of individual factors in the multi-variate analysis?

Minor comments:
1. Instead of "three types of lncRNAs", the authors could write "three lncRNAs".
2. In the abstract: "Three kinds of lncRNAs are involved in the regulation of disease by affecting cytokines and the immune system." : this should not be definitive as further validations are needed to support this statement. Rather the authors can write "The enrichment study indicates that these three lncRNAs are involved in the regulation of disease by affecting cytokines and the immune system".

Validity of the findings

The study supports the conclusion. But some clarifications are still needed (as stated above) before the manuscript can be accepted for publication.

Additional comments

The study aims to find out the roles of lncRNAs as part of putative competing endogenous RNA network in tongue squamous cell carcinoma. While the overall design of the study is well defined and the results support the conclusions, I would like to point out some aspects where the manuscript can be more clear.
1. In the Methods section, proper description of the TCGA dataset should be given. Instead of "RNA sequences", the authors should mention whether they used RNA-Seq v2 and/or miRNA-seq dataset from TCGA, and in which format the RNA-seq data was downloaded; was it raw count or RSEM or FPKM? Also, the authors mentioned "Before conducting differential expression analysis, we processed the data and deleted the RNA and miRNA data that were not expressed"; was there a cut-off for expression or number of samples (e.g. remove genes with < 1 FPKM in 90% of samples)?
2. They mentioned about multi-variate analysis with mRNAs in the ceRNA network. But no figures or tables are included that shows the result of the multi-variate analysis. How was the risk scores and what was the significance of individual factors in the multi-variate analysis?

Minor comments:
1. Instead of "three types of lncRNAs", the authors could write "three lncRNAs".
2. In the abstract: "Three kinds of lncRNAs are involved in the regulation of disease by affecting cytokines and the immune system." : this should not be definitive as further validations are needed to support this statement. Rather the authors can write "The enrichment study indicates that these three lncRNAs are involved in the regulation of disease by affecting cytokines and the immune system".
3. Figure 1, 2, and 3 can be combined into a single figure with multiple panels. The volcano plots can be moved to supplementary figures. Figure 4 is redundant and can be moved into supplementary. Also, Figure 7, 8 and 9 can be combined into a single figure with multiple panels.

Reviewer 2 ·

Basic reporting

Major reviews:
The authors could update the reference literature. For example, in line 57 when they mention oral carcinomas the latest reference was published in 2012. This recomendation is for the lncRNAs too. During the introduction they spent a lot of time describing the mechanism of ceRNA, which is very important. However I would suggest to reduce without losing contend about the mechanism and highlight more the studies that link lncRNAs and tumorigenesis. I would recommend to rearrange the introduction to stay straight to hypothesis. Otherwise it remains fuzzy and the propose is not clear in the end.
Minor reviews:
Regarding the figures 1,2, and 3, please clarify what rows and columns represent. And it would be benefic if the authors put color bars according to the diagnosis of the sample, it would become clearer if these differentially expressed transcripts are able to group samples according to their diagnosis.
Improve the resolution of figures with GSEA analysis, it is really hard to see something.
Tables would be great if each gene are repsented in a line with the name description together.

Experimental design

Major review:
It would be great if the authors describe better how they proceed in the analysis. For example, they used TCGA samples, what kind of files raw sequencing data, processed samples.
1-) The authors performed some clustering analysis or used MDS (multidimensinal scale) to check outliers samples? If they don't I strongly recommend perform that.
2-) Why do the authors used the ENsembl annotation and not the GENCODE annotation once they are working with noncoding RNA.
3-) They check for technical bias in the expression data? If they don't I strongly recommend perform that. And if there is some of these they would have to correct it.
4-) When performing the model of differentially expressed genes, the interest variable was the diganosis, and what about the ajusted variables like sex, age and other? This would make a huge difference in this model .
5-) Perform cut offs using Fold change is not a good strategy, unless the authors show the expression value. For example, a Fold change of 2 in genes that have average expression of 5, would require that control and case have 5 and 10 respectively, however for a gene with average expression of 1000, it would requiere that expression should be 1000 and 2000. In this last case a Fold Change of 1.5 would requiere an average expression of 1500 and 1000, which is very meaningful.
6-) It would be great have a table with clinical variables, just to be aware if the dataset has some biological bias before starting the differentially expression analysis.
7-) In the netwrok analysis, what are the parameters used to measure the binding between miRNA and lncRNAs, what are free energy of binding, or there multiple bindings? If they only used what was described in the literature, it would should be clearer, and would be great knowning how large is this database to make a comparison if what they found is a good result or not.
8-) To perform the network the authors used a correlation measure? Because it is not enough to have the binding, but you have to know the dynamic as well. So I strongly suggest to perform a correlation analysis using expression data and then build the network.
9-) In figure 6, what are the criteria to define high and low expression?
It would be great to know the criteria of these, and show what is the rank of expression value of these RNAs according to each class others transcripts.

Validity of the findings

The results are meaningful, once this study reveals new potential targets and describe mechanisms that could lead to the tongue squamous cell carcinoma. The discussion is well structure and it easy to follow. The findings are useful in tha lncRNA area, as well as in cancer area. The conclusions are sticked to the point.

Additional comments

The main findings of these paper are of interest of scientific community and should be considered for publication. My main concern is about the technical questions, which could change the final results, however the main core of this study, probably, would not change. The disussion and conclusion are easy to follow and if the introduction and results were ajusted in that way would be great. Recommendations about figure and tables were already describe before.

---

## Round 0.2 · Minor Revisions

The manuscript, presenting analysis regulatory network constructed based on long noncoding RNA, has been significantly improved by the authors. The authors are encouraged to involve a technical editor at this stage to improve the quality of written English. There are just some examples of typos:

(ceRNA)regulatory->(ceRNA) regulatory
indicatea poor->indicate poor

In some cases, researcher's name is not necessary, since the citation is given at the end of the sentence, and all other researchers are not mentioned by names. What is the importance of this particular researcher? May be, the name needs to be removed:

Xintang Lan showed -> it has been shown

"We used the R survival package for survival analysis". Did you use the Package ‘survival’? In this case, it should be written "We used the 'survival' package in R for survival analysis", for clarity.

Statistical analysis: Please explain the choice of cut-offs in the GSEA, FDR< 25%. It is much higher threshold than FDR used in other parts of the manuscript.

Reviewer 1 ·

Basic reporting

The authors have addressed all my concerns. But in many places the English should be checked carefully and corrected.

Experimental design

No further comments

Validity of the findings

No further comments

---

## Round 0.3 · accepted · Accept

Thank you very much for fast response and correction of all identified minor issues. Your paper is now suitable for the publication based on the scientific content. Although Accepted, you may still want to further improve the English language via a professional editor, to make sure that the quality of written English is equally sound.

#